# A Channel Imbalance Calibration Scheme with Distributed Targets for Circular Quad-Polarization SAR with Reciprocal Crosstalk

**Xingjie Zhao [1,2]**, **Yunkai Deng [1]**, **Heng Zhang [1]** and **Xiuqing Liu [1,*]**

1 The Department of Space Microwave Remote Sensing System, Aerospace Information Research Institute, Chinese Academy of Sciences, Beijing 100190, China
2 The School of Electronic, Electrical and Communication Engineering, University of Chinese Academy of Sciences, Beijing 100049, China
* Correspondence: lucia@mail.ie.ac.cn

**Abstract:** As polarimetric antennas can be isolated through excellent electronic frameworks in circular quad-polarization (CQP) synthetic aperture radar (SAR) systems, cross-polarization (x-pol) and co-polarization (co-pol) channel imbalances are more challenging and essential to calibrate than crosstalk in polarimetric calibration (PolCAL). In uncalibrated CQP SAR images without corner reflectors (CRs), the reciprocity and reflection symmetry assumptions of the distributed targets are commonly used to estimate the x-pol and co-pol channel imbalances, respectively. To suppress the influence of additive noise on determining channel imbalances through distributed targets, high signal-to-noise ratio (SNR) distributed targets should be obtained from the x-pol and co-pol channels of the CQP SAR images: namely, surface-dominated and volume-dominated targets. However, some reflection symmetry assumptions used in the existing calibration literature have poor applicability with volume-dominated targets, resulting in unsatisfactory estimation results for the co-pol channel imbalance phase. In this paper, we assess the priority of the reflection symmetry properties of volume-dominated targets used to calibrate the co-pol channel imbalance phase in CQP SAR data synthesized from linear quad-polarization data of ALOS, GF-3, and RADARSAT-2. In the theoretical part, high-priority reflection symmetry (termed semireflection symmetry) assumptions are confirmed as the most suitable for estimating the co-pol channel imbalance phase, and were selected to develop an algorithm for estimating the co-pol channel imbalance phase. Furthermore, based on the novel method for estimating the co-pol channel imbalance phase, a channel imbalance calibration scheme is proposed for CQP SAR systems with reciprocal crosstalk, including extracting surface-dominated and volume-dominated targets, and estimating and filtering channel imbalances. We demonstrate the effectiveness of our proposed scheme with CRs in simulated CQP SAR images. The experimental results show that the calibration scheme is an effective workflow for estimating channel imbalances in CQP SAR systems with reciprocal crosstalk.

**Keywords:** circular quad-polarization (CQP); synthetic aperture radar (SAR); channel imbalance calibration scheme; reciprocal crosstalk





## 1. Introduction

Quad-polarization synthetic aperture radar (SAR) systems actively transmit and receive different combinations of electromagnetic waves, including (1) linear quad-polarization (LQP) SAR, which alternately transmits and receives horizontal (H) and vertical (V) electromagnetic waves, (2) circular quad-polarization (CQP) SAR, which alternately transmits and receives left-handed (L) and right-handed (R) electromagnetic waves, and (3) hybrid quad-polarization (HQP) SAR, which transmits L and R electromagnetic waves, according to combinations of different phases of H and V electromagnetic waves, and alternately

receives H and V waves [1,2]. Due to its high sensitivity to clutter backscatter, quad-polarization SAR provides high-quality data for a wide range of remote applications [3–7]. However, hardware system distortions can lead to unfaithful representations of clutter backscatter. It is very important to calibrate the observation matrix to the undistorted measurement, which is known as polarimetric calibration (PolCAL) [8–10]. Polarimetric distortions, which are SAR system parameters, mainly include crosstalk, co-polarization (co-pol) channel imbalance, and cross-polarization (x-pol) channel imbalance [11–15]. For decades, SAR antennas have been highly isolated (better than 32 dB [16–18]), with crosstalk that is stable at different incidence angles. Therefore, x-pol and co-pol channel imbalances in PolCAL merit further study.

　　Although CQP SAR systems have been extensively studied in the imaging radar literature [19], channel imbalance calibration on CQP SAR images has received less attention. Based on the antenna transmit model s-m-a (which minimizes the potential for crosstalk, by ensuring complete physical isolation between channels) and the corresponding receive model, [20] proposed an iterative method to calibrate crosstalk and channel imbalances, using corner reflector (CR) groups (including trihedral, dihedral, and gridded trihedral CRs) in CQP SAR systems with reciprocal crosstalk. In addition, considering that CRs can be used only to determine the distortion of CQP SAR data at specific incidence angles, the reflection symmetry and reciprocity of the distributed targets were also considered when estimating the co-pol and x-pol channel imbalances. As the reciprocity of polarimetric SAR images is generally accepted in polarimetric applications [19,21], and the estimation of the x-pol channel imbalance with reciprocally distributed targets is not perturbed by the reciprocal crosstalk in the CQP SAR calibration, the initial estimate of the x-pol channel imbalance was equal to the true value [20]. However, the statistical reflection symmetry characteristics exhibited by distributed targets in the undistorted CQP SAR images did not fully match the ideal reflection symmetry assumptions, which limited the calibration accuracy of the co-pol channel imbalance. Compared with the CR calibration results, the co-pol channel imbalance phase estimated according to the reflection symmetry properties of the distributed targets differed by 14 degrees ([20], Table 2).In existing LQP systems, the acceptable error of the calibrated co-pol channel imbalance phase is generally within $\pm 10$ degrees ([22], Table 4).

　　The first contribution of this paper is to propose a calibration algorithm for the co-pol channel imbalance phase based on high-priority reflection symmetry (termed semireflection symmetry) assumptions. As additive noise may affect the statistical polarization characteristics of the distributed targets [18,23], high signal-to-noise ratio (SNR) targets in the CQP SAR co-pol channel—namely, volume-dominated targets—should be applied, to calibrate the co-pol channel imbalance. One reflection symmetry condition of LQP SAR images is that the average polarization orientation angle is zero, which can be difficult to achieve for natural targets with large terrain fluctuations and complex scattering characteristics, such as sparse woodland areas [24,25]. Considering that the reflection symmetry of CQP SAR images can be obtained from LQP SAR images through polarimetric basis transformations, the calibration accuracy cannot be guaranteed if the low-priority reflection symmetry properties of volume-dominated targets are used to address the co-pol channel imbalance phase. Therefore, this paper determines the priority of reflection symmetry assumptions in volume-dominated targets with CQP SAR data, and then confirms that semireflection symmetry assumptions are the most suitable for estimating the co-channel imbalance phase. In addition, a calibration method for the co-pol channel imbalance phase is proposed, based on semireflection symmetry properties.

　　The second contribution of this paper is to provide a channel imbalance calibration scheme for CQP SAR systems with reciprocal crosstalk. Firstly, the equivalent number of looks (ENL) and the power ratio of the co-pol and x-pol channels are utilized, to extract volume-dominated and surface-dominated target regions in uncalibrated CQP SAR images. Secondly, the calibration method, via clutter [20], is introduced, to estimate the x-pol channel imbalance according to the surface-dominated targets, under the condition of reciprocal

CQP SAR images and crosstalk. Thirdly, the semireflection symmetry assumptions are implemented, to remove the co-pol channel imbalance. Filter operations are also carried out in the second and third steps, to acquire more robust results than the initial estimates.

The effectiveness of the proposed scheme for calibrating channel imbalances is demonstrated with undistorted CQP SAR images synthesized from LQP SAR images, which were acquired from the Yudaokou airborne flight experiment of the Aerospace Information Research Institute, Chinese Academy of Sciences (AIRCAS). The experiments in this paper show that the proposed scheme can be used to estimate channel imbalances with distributed targets in CQP SAR systems with reciprocal crosstalk. In addition, the proposed method is an effective tool for improving the calibration accuracy of the co-pol channel imbalance phase.

The remainder of this paper is organized as follows. Section 2 introduces the basis of CQP SAR PolCAL, including the distortion model and the priority of the reflection symmetry assumptions. The channel imbalance calibration scheme is presented in Section 3. Section 4 introduces the experimental approach. Some artificially adjusted parameters, and the crosstalk tolerance of the proposed scheme, are discussed in Section 5. Finally, Section 6 provides the conclusions of this paper, and discusses future work.

## 2. Basis of CQP SAR PolCAL

### 2.1. Distortion Model

In CQP SAR PolCAL, the undistorted backscatter matrix $[S]$ relates the observed backscatter matrix $[O]$ to the transmit distortion matrix $[T]$, the receive distortion matrix $[R]$, and the random additive thermal noise $[N]$, according to the following distortion model [20]:

$$
\begin{bmatrix} O_{ll} & O_{lr} \\ O_{rl} & O_{rr} \end{bmatrix} = \begin{bmatrix} r_{ll} & r_{lr} \\ r_{rl} & r_{rr} \end{bmatrix} \begin{bmatrix} S_{ll} & S_{lr} \\ S_{rl} & S_{rr} \end{bmatrix} \begin{bmatrix} t_{ll} & t_{lr} \\ t_{rl} & t_{rr} \end{bmatrix} + \begin{bmatrix} N_{ll} & N_{lr} \\ N_{rl} & N_{rr} \end{bmatrix}
$$
$$
\Leftrightarrow [O] = [R][S][T] + [N]
$$
(1)

where the subscripts $l$ or $r$ indicate left-handed or right-handed electromagnetic waves. $S_{pq}$ or $O_{pq}$ ($p$ or $q = l$ or $r$, similarly hereinafter) denote the undistorted or distorted complex scattering factor transmitted by antenna $q$ and received by antenna $p$; $t_{pq}$ denotes the product of the common amplifier factor at antenna $q$, and the scaling factor when antenna $q$ transmits polarization $p$; $r_{pq}$ denotes the product of the common amplifier factor at antenna $p$, and the scaling factor when antenna $p$ receives polarization $q$; $n_{pq}$ denotes the additive noise in the receive channel, $p$, when transmitting $q$. Without loss of generality, we assume that the target backscattering and additive noise are uncorrelated, and that the components in $[N]$ are independent of one another [20,24]. Considering that the Faraday rotation angle can be well-estimated by external reference [22,24], we do not consider the impact of the Faraday rotation angle on (1). Row-based factorization is applied, to transform $[O]$, $[S]$, and $[N]$ to $[O_{4LR}]$, $[S_{4LR}]$, and $[N_{4LR}]$:

$$
\begin{bmatrix} O_{ll} \\ O_{lr} \\ O_{rl} \\ O_{rr} \end{bmatrix} = Y \begin{bmatrix} k^2 & 0 & 0 & 0 \\ 0 & k & 0 & 0 \\ 0 & 0 & k & 0 \\ 0 & 0 & 0 & 1 \end{bmatrix} \begin{bmatrix} \alpha & 0 & 0 & 0 \\ 0 & 1 & 0 & 0 \\ 0 & 0 & \alpha & 0 \\ 0 & 0 & 0 & 1 \end{bmatrix} \begin{bmatrix} 1 & v & w & vw \\ z & 1 & zw & w \\ u & uv & 1 & v \\ uz & u & z & 1 \end{bmatrix} \begin{bmatrix} S_{ll} \\ S_{lr} \\ S_{rl} \\ S_{rr} \end{bmatrix} + \begin{bmatrix} N_{ll} \\ N_{lr} \\ N_{rl} \\ N_{rr} \end{bmatrix}
$$
$$
\Leftrightarrow [O_{4LR}] = Y[K_4][Q][X][S_{4LR}] + [N_{4LR}]
$$
(2)

where $k$, $\alpha$, and $u/v/w/z$ were the co-pol channel imbalance, x-pol channel imbalance, and crosstalk components, respectively; $[K_4]$, $[Q]$, and $[X]$ are the corresponding co-pol channel imbalance, x-pol channel imbalance, and crosstalk matrices, respectively; and $Y$ represents the absolute calibration factor. The relationship between (1) and (2) can be expressed as

$$
Y = r_{rr}t_{rr}, k = r_{ll}/r_{rr}, \alpha = \frac{t_{ll}/t_{rr}}{r_{ll}/r_{rr}}, u = r_{rl}/r_{rr}, v = t_{rl}/t_{ll}, w = r_{lr}/r_{ll}, z = t_{lr}/t_{rr}
$$
(3)

In PolCAL, $Y$ is usually considered to not impair the relative relationship of polarimetric channels [20,22,24,26]; therefore, the calibration process of $Y$ is not considered in this article. The coherence information of each polarimetric channel can be adequately described by CQP SAR systems, which provide constraints for distributed target calibration. The corresponding covariance matrix (2) can be written as

$$[M_{4\text{LR}}] = [K_4][Q][X][C_{4\text{LR}}]([K_4][Q][X])^{\dagger} + [C_n] \tag{4}$$

where $[M_{4\text{LR}}]$, $[C_{4\text{LR}}]$, and $[C_n]$ are the covariance matrices corresponding to $[O_{4\text{LR}}]$, $[S_{4\text{LR}}]$, and $[N_{4\text{LR}}]$, respectively, and the superscript $\dagger$ represents the conjugate and transpose matrix operations. It should be stressed that $[C_n]$ is a four-by-four diagonal matrix, in which the first and second elements are equal, and the third and fourth elements are equal [20]. As multiplication of diagonal matrices is commutative, (4) can be transformed as

$$[M_{4\text{LR}}] = [Q][K_4][X][C_{4\text{LR}}]([Q][K_4][X])^{\dagger} + [C_n] \tag{5}$$

In this paper, (2) and (5) are utilized to solve the channel imbalances with reciprocal crosstalk. When reciprocal crosstalk and noise are ignored, the x-pol and co-pol channel imbalances can be estimated by the reciprocity and reflection symmetry of distributed targets [20], which can be written as:

$$|\alpha| = \sqrt{M_{33}/M_{22}} \tag{6}$$

$$\angle(\alpha) = \angle(M_{32}) \tag{7}$$

$$|k| = \sqrt[4]{\frac{M_{11}/M_{44}}{|\alpha|^2}} \tag{8}$$

$$\angle(k) = \frac{1}{2}\angle(-M_{13}/M_{42}) + n\pi(n = 0, \pm1, \pm2, \cdots) \tag{9}$$

$$\angle(k) = \frac{1}{2}\angle\left(-\frac{M_{12}/M_{43}}{\alpha/\alpha^*}\right) + n\pi(n = 0, \pm1, \pm2, \cdots) \tag{10}$$

$$\angle(k) = \frac{1}{2}\angle\left(\frac{M_{14}}{\alpha}\right) + \frac{1 - \text{sgn}(C_{14})}{2} + n\pi(n = 0, \pm1, \pm2, \cdots) \tag{11}$$

where $\angle(\cdot)$ and $|\cdot|$ represent the phase and amplitude, respectively, of a complex number; $M_{ij}$ and $C_{ij}$ are the ith row and jth column components of $[M_{4\text{LR}}]$ and $[C_{4\text{LR}}]$; and the superscript $*$ indicates the conjugate operator.

### 2.2. Priority Verification of Reflection Symmetry in CQP SAR Images

The distributed targets with reflection symmetry are composed of single targets with mean-zero helicity and mean-zero orientation [9,27]. When the repeated equations, due to the reciprocity of the CQP SAR images, are removed, the reflection symmetry assumptions of the distributed targets can be expressed as

$$\begin{cases} \text{Lin}_1 : \langle S_{hh}S_{hv}^* \rangle = 0 & \\ \text{Lin}_2 : \langle S_{vv}S_{hv}^* \rangle = 0 & \text{Line Basis} \\ \text{Cir}_1 : \langle S_{ll}S_{ll}^* \rangle - \langle S_{rr}S_{rr}^* \rangle = 0 & \\ \text{Cir}_2 : \Im(\langle S_{ll}S_{rr}^* \rangle) = 0 & \\ \text{Cir}_3 : \langle S_{lr}^*(S_{ll} + S_{rr}) \rangle = 0 & \text{Circular Basis} \end{cases} \tag{12}$$

where $\langle\cdot\rangle$ is the spatial average operator of a matrix, which is applied to remove coherent speckle noise; $\Im(\cdot)$ denotes the imaginary operation; and $\text{Lin}_i/\text{Cir}_j$ denote the $i/j$th

reflection symmetry assumption of the LQP/CQP SAR images. The circular basis in (12) can be obtained using a unitary basis-change matrix $[U_{lr}]$, via a unitary consimilarity transformation [28], as follows:

$$[S_{4LR}] = \left( \left( \frac{1}{\sqrt{2}} \begin{bmatrix} 1 & j \\ j & 1 \end{bmatrix} \right) \otimes \left( \frac{1}{\sqrt{2}} \begin{bmatrix} 1 & j \\ j & 1 \end{bmatrix} \right) \right)^T \begin{bmatrix} S_{hh} & S_{hv} & S_{vh} & S_{vv} \end{bmatrix}^T$$

$$\Leftrightarrow [S_{4LR}] = ([U_{lr}] \otimes [U_{lr}])^T [S_{4HV}]$$

(13)

where $\otimes$ represents the Kronecker product of a matrix, and the superscript $T$ represents the transpose operator.

2.2.1. Verification of Reflection Symmetry Used in the *k* Calibration Methods of [20]

In [20], $\text{Cir}_2$ and $\text{Cir}_3$ were used to estimate the phase of *k* through (9) to (11). Next, we rewrite $\text{Cir}_2$ and $\text{Cir}_3$ as $\text{ACir}_2$ and $\text{ACir}_3$:

$$\text{ACir}_2 : \angle(\langle S_{ll} S_{rr}^* \rangle) = n\pi, n = 0, \pm 1, \pm 2 \cdots$$

(14)

$$\text{ACir}_3 : \angle \left( -\frac{\langle S_{ll} S_{lr}^* \rangle}{\langle S_{rr} S_{lr}^* \rangle} \right) = 2n\pi, n = 0, \pm 1, \pm 2 \cdots$$

(15)

to more directly judge whether the reflection symmetry is satisfied in the calibration methods of *k*.

When calibrating different channel imbalances, high SNR targets must be selected in various channels: thus, the calibration results are less affected by additive noise. By analyzing the process of transforming LQP SAR images into CQP SAR images (13), we find that energy-dominated targets in co-pol channels in CQP SAR images exhibit volume-scattering features; therefore, volume-dominated targets should be used to calibrate *k* by applying reflection symmetry. Next, nine LQP SAR images from L-band ALOS, C-band GaoFen-3 (GF-3), and C-band RADARSAT-2 (RD-2) are used to analyze the reflection symmetry of the volume-dominated targets. The specific details include the following steps: (1) the volume-dominated targets are obtained by applying the optimal non-negative eigenvalue decomposition (NNED) [29], which is an effective method of reducing the negative power problem and the overestimated contribution of the volume scattering affected by Freeman—Durden decomposition [30]; (2) $|\text{Cir}_1|/2\text{Span}$, $\text{ACir}_2)$ and $(\text{ACir}_3$ are calculated by (12), (14) and (15), respectively; (3) histograms of $|\text{Cir}_1|/2\text{Span}$ and phase distributions of $\text{ACir}_2$ and $\text{ACir}_3$ are obtained, to confirm the difference between the assumptions exhibited by the actual SAR systems and the ideal theoretical assumptions. The final verification results are shown in Figure 1.

In Figure 1a, we illustrate the $|\text{Cir}_1|/2\text{Span}$ histogram of nine images, where $|\text{Cir}_1|$ values are usually small but nonzero compared to the span value. The $\text{ACir}_2$ and $\text{ACir}_3$ phase distributions of the volume-dominated targets are shown in Figure 1b,c, which are not constant at $0°$ or $\pm180°$, but have approximately equal probability at all angles. Therefore, unsatisfactory *k*-angle calibration results may be obtained if $\text{ACir}_2$ and $\text{ACir}_3$ are used in volume-dominated areas.

2.2.2. $\text{Cir}_3$ Division for Reflection Symmetry Verification

In the common method, $\text{Cir}_3$, which is represented by (12), can be decomposed into an amplitude and a phase, and the phase of $\text{Cir}_3$ is applied, to determine the *k* phase ((9) and (10)). However, the amplitude of $\text{Cir}_3$ also presents some information about the reflection symmetry. In Figure 2, $|\text{Cir}_3|/2\text{Span}$ (solid lines) is investigated in the volume-dominated targets, and $|\text{Cir}_1|/2\text{Span}$ (dotted lines) is also shown, for comparison. Clearly, $\text{Cir}_3$ and $\text{Cir}_1$ have the same order of small amplitude in most volume-dominated targets. Compared with $\text{ACir}_3$, which has a similar distribution across the entire phase domain, $|\text{Cir}_3|$ may be a better choice for calculating the *k* phase. However, when $|\text{Cir}_3|/2\text{Span}$ and

$|Cir_1|/2Span$ are close to 0, the frequency of $|Cir_1|/2Span$ is noticeably greater than that of $|Cir_3|/2Span$. Thus, there may be a better method of calibrating the phase of $k$ than directly using $|Cir_3|$.

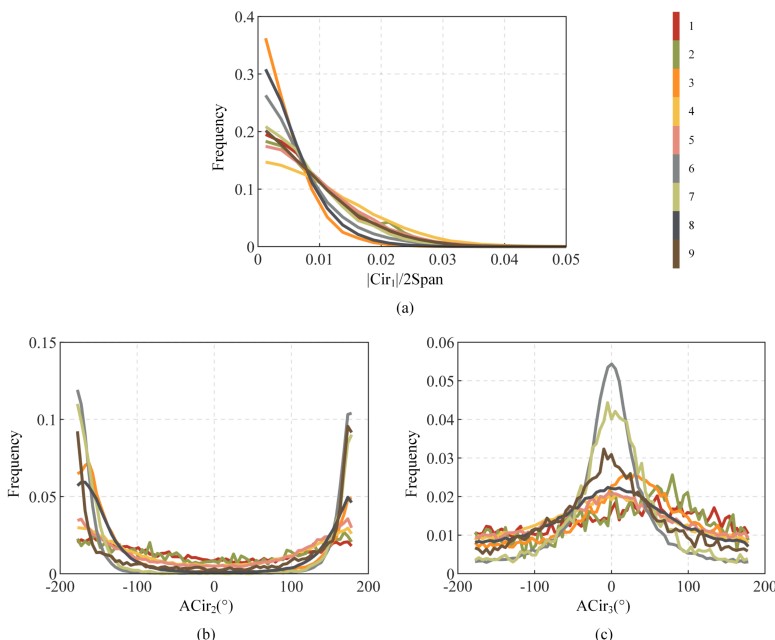

**Figure 1.** Histograms of (**a**) $|Cir_1|/2Span$, (**b**) $ACir_2$, and (**c**) $ACir_3$ of volume-dominated targets in the nine CQP SAR images.

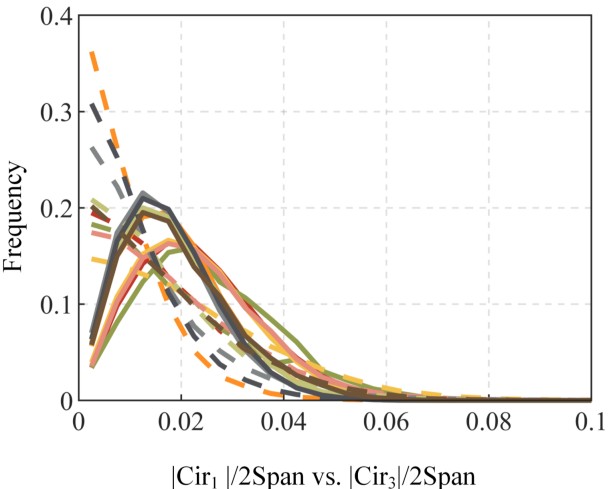

**Figure 2.** Histograms of $|Cir_1|/2Span$ (dotted lines) and $|Cir_3|/2Span$ (solid lines).

In addition to dividing a complex number into its amplitude and phase, complex numbers can be decomposed into imaginary and real parts. Compared to $|Cir_3|$, the imaginary and real parts of $Cir_3$ may be less powerful than those of $|Cir_3|$. Thus, $Cir_3$ can be divided into an imaginary part, $Cir_4$, and a real part, $Cir_5$, to achieve a more accurate description of the reflection symmetry in the volume-dominated targets.

The histograms of $|Cir_4|/2Span$ (dashed line) and $|Cir_5|/2Span$ (solid line) are shown in Figure 3a. Obviously, in the volume-dominated area, $|Cir_4|/2Span$ and $|Cir_5|/2Span$ have the same number level of points close to 0, and more points close to 0 than $|Cir_3|/2Span$. Furthermore, the percentages of all volume-dominated points with $|Cir_4|/2Span < 0.1$ and $|Cir_5|/2Span < 0.1$ in each image are plotted in Figure 3b. Both have high percentages above 98% in all nine images. Additionally, the percentage of $|Cir_4|/2Span < 0.1$ is higher

than that of $|\text{Cir}_5|/2\text{Span} < 0.1$. Thus, the simulation experiments with the undistorted CQP SAR images show that $|\text{Cir}_4|$ and $|\text{Cir}_5|$ are closer to the ideal reflection symmetry assumptions than $\text{ACir}_2$, $\text{ACir}_2$, or $|\text{Cir}_3|$. Additionally, while the value of $|\text{Cir}_4|$ near 0 is slightly higher than that of $|\text{Cir}_5|$ in the undistorted CQP SAR images, the difference is small.

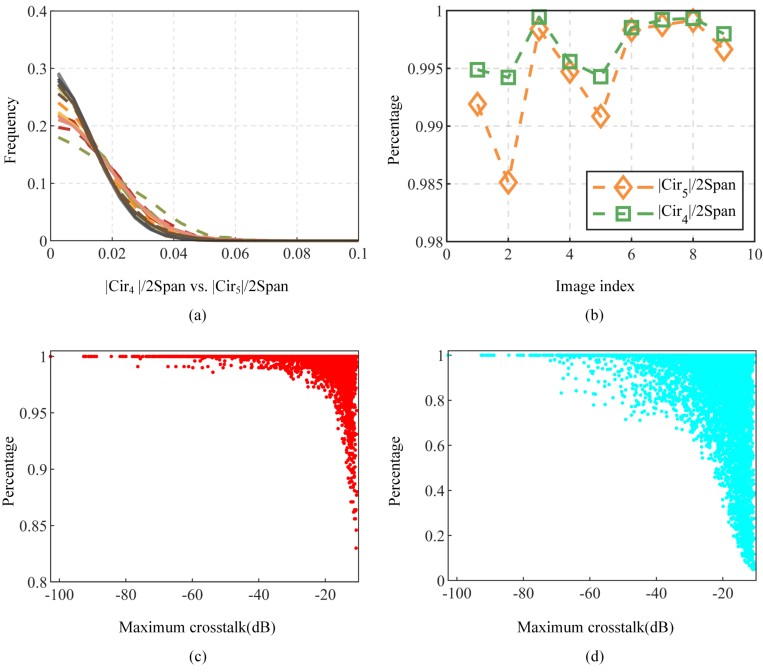

**Figure 3.** Comparison of $|\text{Cir}_4|/2\text{Span}$ and $|\text{Cir}_5|/2\text{Span}$: (**a**) histograms of $|\text{Cir}_4|/2\text{Span}$ and $|\text{Cir}_5|/2\text{Span}$ in the volume-dominated area of nine undistorted CQP SAR images; (**b**) percentages of all volume-dominated points with $|\text{Cir}_4|/2\text{Span} < 0.1$ and $|\text{Cir}_5|/2\text{Span} < 0.1$ in each undistorted CQP SAR image; (**c**,**d**) percentages of all reflection symmetry points with $|\text{Cir}_4|/2\text{Span} < 0.1$ and $|\text{Cir}_5|/2\text{Span} < 0.1$ in the presence of reciprocal crosstalk.

Based on the above analysis, $\text{Cir}_4$ and $\text{Cir}_5$ are more suitable for estimating the phase of $k$ by analyzing the prioritized reflection symmetry in undistorted volume-dominated areas. Furthermore, the priority between $\text{Cir}_4$ and $\text{Cir}_5$ should be determined in the reflection symmetry and reciprocity simulations of CQP SAR data with distortions (i.e., reciprocal crosstalk and $\alpha$), to obtain a higher $k$ phase calibration accuracy. In the following, the influence of $\alpha$ is ignored, because the calibration results of estimating the x-pol channel imbalance with (6) and (7) in the reciprocal targets are the same with true value when the CQP SAR antennas have reciprocal crosstalk [20]. It should be stressed that reciprocity of targets is generally acceptable in polarimetric applications.

For this part, 1000 reflection symmetry points were simulated, to conduct 20,000 Monte Carlo trials. During each trial, we imposed the reciprocal crosstalk unknowns, with random amplitudes from 0 to 0.3 ($0 \leqslant |u| = |z|, |v| = |w| \leqslant 0.3(-10.4576 \text{ dB})$), and random phases ($-180° \leqslant \angle u = \angle z, \angle v = \angle w \leqslant 180°$) were added to the simulated data first of all. Then, the percentages of $|\text{Cir}_4|/2\text{Span} < 0.1$ and $|\text{Cir}_5|/2\text{Span} < 0.1$ were counted in the presence of reciprocal crosstalk. Finally, we verified the priority of $|\text{Cir}_4|/2\text{Span}$ and $|\text{Cir}_5|/2\text{Span}$, according to the maximum crosstalk, ($\{\text{dB}(u), \text{dB}(v), \text{dB}(w), \text{dB}(z)\}_{\text{max}}$).

The results are shown in Figure 3c,d, where the X axis is the maximum added crosstalk, and the Y axis is the percentages of $|\text{Cir}_4|/2\text{Span} < 0.1$ and $|\text{Cir}_5|/2\text{Span} < 0.1$. As the crosstalk increases, an overall decrease can be seen in the ratios of $|\text{Cir}_4|/2\text{Span} < 0.1$ and $|\text{Cir}_5|/2\text{Span} < 0.1$. However, at approximately $-60$ dB, the drop rate and drop value of $|\text{Cir}_5|/2\text{Span}$ are noticeably higher than those of $|\text{Cir}_4|/2\text{Span}$. When the maximum crosstalk increases to approximately $-15$ dB, the percentage of $|\text{Cir}_5|/2\text{Span}$ larger than 0.1 is approximately 0.1 (10%), while the percentage of $|\text{Cir}_4|/2\text{Span}$ larger than 0.1 is

approximately 0.85 (85%). Therefore, $\text{Cir}_4$ is more important than $\text{Cir}_5$, when calibrating the $k$ phase in the actual calibrated and simulated uncalibrated data, and $\text{Cir}_4$ is used to develop an algorithm to estimate the $k$ phase.

## 3. Materials and Methods

Due to the high cost of deploying CRs, it is necessary to use distributed targets to determine the channel imbalances. This section describes the complete scheme of calibrating channel imbalances in CQP SAR images with reciprocal crosstalk. Based on some assumptions of CQP SAR systems and distributed targets, the proposed scheme consists of three main steps. Firstly, ENL and $R_{vb}$ are utilized to extract the surface-dominated and volume-dominated targets as the calibration candidate pixels. Secondly, the reciprocity of the surface-dominated targets is utilized, to determine the x-pol channel imbalance $\alpha$. Thirdly, the high-priority reflection symmetry assumptions of the surface-dominated targets are utilized, to determine the co-pol channel imbalance $k$. In the second and third steps, filter operations are performed, to obtain more robust results. The schematic workflow is shown in Figure 4.

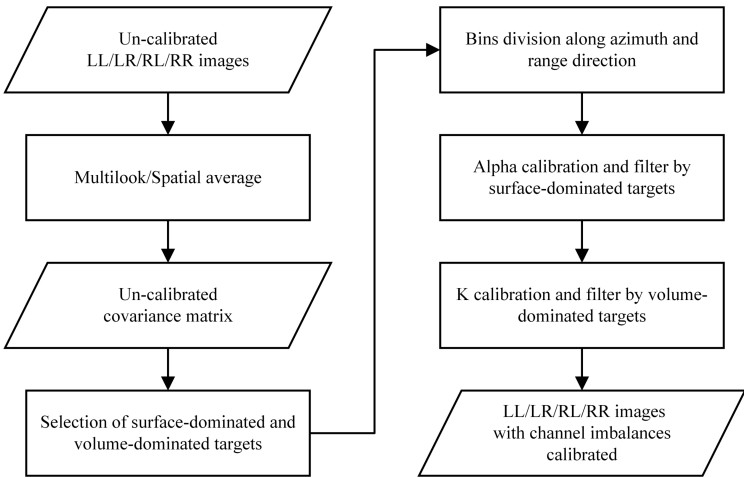

**Figure 4.** Channel imbalance calibration scheme for CQP SAR images with reciprocal crosstalk.

### 3.1. Assumptions of Natural Targets and CQP SAR System

3.1.1. Reciprocity of Natural Targets and Semireflection Symmetry of Volume-Dominated Targets

The first assumption for natural targets is reciprocity, which is a well-recognized assumption that has been applied in many polarization applications [31]. In LQP SAR systems, the reciprocity of natural targets can be expressed as $S_{hv} = S_{vh}$. Combined with (13), the reciprocity of CQP SAR images can be expressed as $S_{lr} = S_{rl}$. As described in Section 2.2, $\text{Cir}_1$ and $\text{Cir}_4$ are statistically derived from CQP SAR data (either undistorted CQP SAR data with reciprocity and reflection symmetry or simulated reciprocal CQP SAR with reciprocal crosstalk and reflection symmetry), to determine important reflection symmetry characteristics in volume-dominated targets. Next, we hypothesize that volume-dominated targets satisfy low $\text{Cir}_1$ and $\text{Cir}_4$ power, which is termed the semireflection symmetry of the volume-dominated targets.

3.1.2. Antenna Reciprocity, No Azimuth Drifting, and Range Drifting

Here, reasonable assumptions are made for CQP SAR systems. Firstly, the antennas are passive and can therefore be assumed to behave reciprocally: that is, the crosstalk during transmission and reception is equal [20]. We can derive $u = z$ and $w = v$ by applying the antenna reciprocity assumption to the calibration model (1).

Secondly, the system distortion changes with the flight time, because the system hardware is unstable. In addition, the unknown antenna distortions are reflected in the

images at different incident angles, [20,32,33]. Considering that the azimuth time of each CQP SAR image is relatively short, we assume here that the system distortion parameters are constant along the azimuth direction: namely, no azimuth drifting. The distortion parameters are considered to be unstable along the range direction: namely, range drifting. According to these criteria, the uncalibrated CQP SAR images can be divided into bins along the azimuth and range directions, and the distortion parameters are consistent in different azimuth bins along each range direction in the following process.

### 3.2. Distributed Targets Selection for PolCAL

Before using PolCAL to determine the channel imbalances, it is important to extract distributed targets with appropriate scattering characteristics, so as to determine the accuracy of PolCAL. One criterion for extracting distributed targets is that ground targets with features that are less affected by additive noise in the corresponding calibration channels should be selected. By analyzing the process of transforming from the LQP SAR to the CQP SAR (13), we find that two co-pol channels, $S_{ll}$ and $S_{rr}$, mainly exhibit volume-scattering features, while the x-pol channels $S_{lr}$ and $S_{rl}$ mainly show surface-scattering features. In natural scattering mechanisms, the volume-dominated target regions are mainly forest areas and complex artificial areas with specific incident angles, while the surface-dominated target regions mainly include bare soil and airport runways. However, identifying different incidence artificial areas is a time-consuming task, which imposes a computational burden on calibration with distributed targets: to address this issue, we select natural targets with strong volume scattering power and strong surface scattering power, after removing complex artificial targets through polarimetric parameters.

It should be pointed out that selecting appropriate distributed targets is challenging, because the scattering properties of the uncalibrated CQP SAR images deviate significantly from the polarization features of the actual undistorted data; therefore, after analyzing many extraction methods in LQP SAR images (such as [24,34,35]), ENL and $R_{vb}$ are introduced, to select the volume-dominated and surface-dominated targets. In general, compared with natural areas, the estimated ENL of complex artificial areas is smaller than the true value [24]; therefore, ENL is selected, to remove the effect of artificial areas. $R_{vb}$—namely,

$$R_{vb} = \frac{\sqrt{|M_{44}M_{11}|}}{|M_{23}|,} \tag{16}$$

which is related to the ratio of co-pol channel power and x-pol channel power—is used in the remaining regions, to select potential volume-dominated and surface-dominated targets. When we neglect the influences of the second-order items of crosstalk values, and the product of additive noise and crosstalk, $R_{vb}$ is virtually unaffected by distortion factors: namely, $\frac{\sqrt{|M_{44}M_{11}|}}{|M_{23}|} \approx \frac{\sqrt{|C_{44}C_{11}|}}{|C_{23}|}$. Larger $R_{vb}$ values are used to select volume-dominated targets, while smaller $R_{vb}$ values are used to select surface-dominated targets. Based on the above description, the areas dominated by volume scattering and surface scattering are filtered according to the following criteria:

$$\begin{cases} ENL \geqslant th_e \to & \begin{matrix} R_{vb} \sim< th_{rl} & surface-dominated \\ R_{vb} \sim> th_{ru} & volume-dominated \end{matrix} \end{cases} \tag{17}$$

After setting the ENL threshold, $th_e$, to discard urban points in the CQP SAR images, the $R_{vb}$ values in the remaining areas are sorted from small to large, and the index of the surface-dominated points ranges from 1 to the total number of points multiplied by $th_{rl}$; the $R_{vb}$ values in the remaining areas are sorted from large to small, and the index of the volume-dominated points ranges from 1 to the total number of points multiplied by $th_{ru}$.

As previously mentioned, it is rather difficult to select scatter-dominated areas in uncalibrated CQP SAR images. In Sections 4 and 5, we demonstrate that the proposed

selection strategy (17) is an effective method for selecting volume-dominated and surface-dominated targets to estimate channel imbalances.

### 3.3. Channel Imbalances Calibration by Surface-Dominated and Volume-Dominated Targets

In the following section, the channel imbalances are calibrated with distributed targets, based on the selected volume-dominated and surface-scattered areas, and the assumptions about natural targets and CQP SAR systems.

**Step 1:** Considering that reciprocal crosstalk and $k$ will not affect the calibration accuracy of $\alpha$ in reciprocal targets, (6) and (7), in [20], are utilized to calibrate $\alpha$ in the reciprocal surface-dominated targets. In the calibration process, a situation may arise in which there is no selected area along a range direction, resulting in an inability to determine the distortion parameters.

Here, we introduce the filtering operations on the amplitude and phase of $\alpha$, to extend the amplitude and phase of the finite $\alpha$ values to the entire range of cells. Meanwhile, the filter operations can eliminate the errors caused by the selection of improper scattering areas, to a large extent, to obtain robust results.

The filtering operations can be divided into three steps: (1) we first obtain the $\alpha$ solution corresponding to the extracted surface-dominated areas in the uncalibrated CQP SAR images; (2) considering that the phases and amplitudes of $\alpha$ values obtained at different azimuths along a range direction are approximately the same, according to the no azimuth drift property, we perform first-order polynomial fitting on the azimuths with these values, and the fitting results are averaged to determine the amplitude and phase of $\alpha$ in this range direction; (3) the bin index (horizontal axis)-phase/amplitude of the $\alpha$ (vertical axis) coordinate system is rotated, to determine the main gradient direction, and the histogram statistics of the rotated data are obtained. A total of 70% of the points near the highest value in the histogram are selected for first-order polynomial fitting, to derive the continuous phases and amplitudes of $\alpha$ in the whole CQP SAR image.

For each bin, the corresponding filtered $\alpha$ values are used to calibrate the covariance matrices of all volume-dominated pixels, as follows:

$$\left[M'_{4LR}\right] = \left([Q]\right)^{-1}[M_{4LR}]\left([Q]^{\dagger}\right)^{-1} = [K_4][C_{4LR}][K_4]^{\dagger} \tag{18}$$

**Step 2:** Semireflection symmetry is used to estimate $k$ by the following process. By expanding (18), $\text{Cir}_1$ and $\text{Cir}_4$ can be expressed as

$$\text{Cir}_1 : |p|^4 M'_{11} - M'_{44} = 0 \tag{19}$$

$$\text{Cir}_4 : \Im\left(|p|^2 p M'_{12} + p M'_{24}\right) = 0 \tag{20}$$

where $p$ is the inverse of $k$. The amplitude of $p$ can be easily obtained from (19):

$$|p| = \sqrt[4]{\frac{M'_{44}}{M'_{11}}} \tag{21}$$

Compared to the amplitude of $p$, the phase of $p$ is more difficult to determine accurately. Next, we propose an algorithm based on (20), to estimate the phase of $p$. We find that $p = 0$ is a constant solution of (20), which is inconsistent with the actual situation. Therefore, we divide $|p|$ by (20), to obtain $\text{UCir}_4$:

$$\text{UCir}_4 : \Im\left(|p| p M'_{12} + \frac{p}{|p|} M'_{24}\right) = 0 \tag{22}$$

Next, Newton's iterative method is utilized, to solve (22). Assuming that $p = a + b\mathrm{j}$, where $a$ and $b$ are real numbers, the associated differentials of $\mathrm{UCir}_4$ are

$$\begin{cases} \partial_a(\mathrm{UCir}_4) = -\frac{ab\Re(M'_{24})}{(a^2+b^2)^{\frac{3}{2}}} + \sqrt{a^2+b^2}\Im(M'_{12}) + \frac{a(a\Im(M'_{12})+b\Re(M'_{12}))}{\sqrt{a^2+b^2}} - \frac{a^2\Im(M'_{24})}{(a^2+b^2)^{\frac{3}{2}}} + \frac{\Im(M'_{24})}{\sqrt{a^2+b^2}} \\ \partial_b(\mathrm{UCir}_4) = -\frac{ab\Im(M'_{24})}{(a^2+b^2)^{\frac{3}{2}}} + \sqrt{a^2+b^2}\Re(M'_{12}) + \frac{b(a\Im(M'_{12})+b\Re(M'_{12}))}{\sqrt{a^2+b^2}} - \frac{b^2\Re(M'_{24})}{(a^2+b^2)^{\frac{3}{2}}} + \frac{\Re(M'_{24})}{\sqrt{a^2+b^2}} \end{cases} \quad (23)$$

where $\Re(\cdot)$ denotes the real operation of a complex number. Considering that at least two sets of data are needed to determine the two unknowns in (23), several azimuth bins at the same range are used, to obtain unique $p$ values in the corresponding range bin, according to the azimuth non-drifting property. Assuming that the selected volume-dominated areas are in $D(D \geqslant 2)$ azimuth bins, in the $d$ iteration, the relationship between $p$ and the changing values of $a$ and $b$—namely, $\Delta a$ and $\Delta b$—can be expressed as

$$p_d = p_{d-1} + \Delta a + j\Delta b \tag{24}$$

where

$$[\Delta a, \Delta b]^T = \left( [\partial(\mathrm{UCir}_4)]^T [\partial(\mathrm{UCir}_4)] \right)^{-1} [\partial(\mathrm{UCir}_4)]^T [\mathrm{UCir}_4] \tag{25}$$

$$[\partial(\mathrm{UCir}_4)] = \begin{bmatrix} \partial_a(\mathrm{UCir}_4)_1 & \partial_a(\mathrm{UCir}_4)_2 & \cdots & \partial_a(UCir_4)_D \\ \partial_b(\mathrm{UCir}_4)_1 & \partial_b(\mathrm{UCir}_4)_2 & \cdots & \partial_b(UCir_4)_D \end{bmatrix}^T \tag{26}$$

$$[\mathrm{UCir}_4] = \begin{bmatrix} -(\mathrm{UCir}_4)_1 & -(\mathrm{UCir}_4)_2 & \cdots & -(\mathrm{UCir}_4)_D \end{bmatrix}^T \tag{27}$$

The initial $p$ values are set over a wide range, to minimize the effect on the Newton iteration method. The amplitudes are set at $-3$ dB$\sim$3 dB, and the phases are set at $-180°\sim180°$. The final $p$ value is selected according to the minimum residual error, which can be calculated by

$$\mathrm{error}_{re} = \left\| [\partial(\mathrm{UCir}_4)][\Delta a, \Delta b]^T - [\mathrm{UCir}_4] \right\|_2 \tag{28}$$

where $\|\cdot\|_2$ denotes the 2-norm of the matrix. Note that $p$ and $p * e^{\pm j\pi}$ are the solutions of (22), and further processing is required to eliminate ambiguous values.

Combining the above solutions, based on $\mathrm{Cir}_1$ and $\mathrm{UCir}_4$, (19) can be used to acquire only the amplitude of $p$, while (20) can be applied to determine the amplitude and phase of $p$, but the final phases have $\pm\pi$ ambiguity. Next, we define the filtering operations of $k$, which are roughly the same as the magnitude and phase filtering of $\alpha$. The differences can be summarized as follows: (1) in the first amplitude filtering step, the initial value is the amplitude obtained by combining two amplitude solutions of $\mathrm{Cir}_1$ and $\mathrm{UCir}_4$; (2) in the phase filtering operation of $k$, the difference lies in the third step: after deriving the histogram statistics of the rotated coordinates, according to the main gradient, the histogram can be divided into several parts, because of the $\pm\pi$ phase shift. In each part, 80% of the points near the peak are selected to perform first-order polynomial fitting. The fitting results of the $\pm\pi$ shift phases and amplitudes are applied to calibrate $k$, and the correlation between the orientation angles obtained by the polarization data and the digital elevation model (DEM) decreases [22]. Therefore, the correct phase can be obtained according to the correlation. If the DEM cannot be obtained in the selected area, only the angle fitted by the data in the middle part of the histogram is selected as the final solution.

Finally, the obtained solutions of the phase and amplitude of $k$ are introduced into the calibration matrix through (18), to obtain the final calibrated CQP SAR images.

## 4. Experiments and Results

In September 2021, AIRCAS carried out L-band airborne LQP SAR flight experiments at Yudaokou Airport, Hebei Province, China. The main SAR system characters are shown in Table 1. The typical features of the ground targets included farmland, forests, weedy

fields, and airport runways. The original data size was $2000 \times 2000$ pixels in the azimuth and range directions, and the resolution was $0.659 \times 0.681$ m. Several kinds of CRs (see Figure 5) were deployed, to perform different scientific tasks, including trihedral CRs, $0°$ dihedral CRs, and $22.5°$ rotated dihedral CRs. In this section, the L-band CQP SAR data, which are synthesized according to the reciprocal LQP SAR data, are used to estimate the channel imbalances. We add different reciprocal crosstalk and channel imbalances to the reciprocal CQP SAR data, to simulate uncalibrated CQP SAR data. A $7 \times 7$ averaging window is used, to estimate the covariance matrix. Because the additional noise levels of different SAR systems vary, the following experiments are divided into two parts, to validate the robustness and accuracy of the proposed scheme: (1) channel imbalance calibration without additionally additive noise, and (2) channel imbalance calibration with additionally additive noise.

**Table 1.** LQP SAR system technical specifications.

| Characters | Value |
|---|---|
| Center Frequency (GHz) | 1.26 |
| Bandwidth (MHz) | 200 |
| Pulse Repeat Frequency (Hz) | 1805 |
| Flight Hight (m) | 4645.64 |
| Platform Velocity (m/s) | 84.93 |
| Incidence Angle (Degree) | 55 |

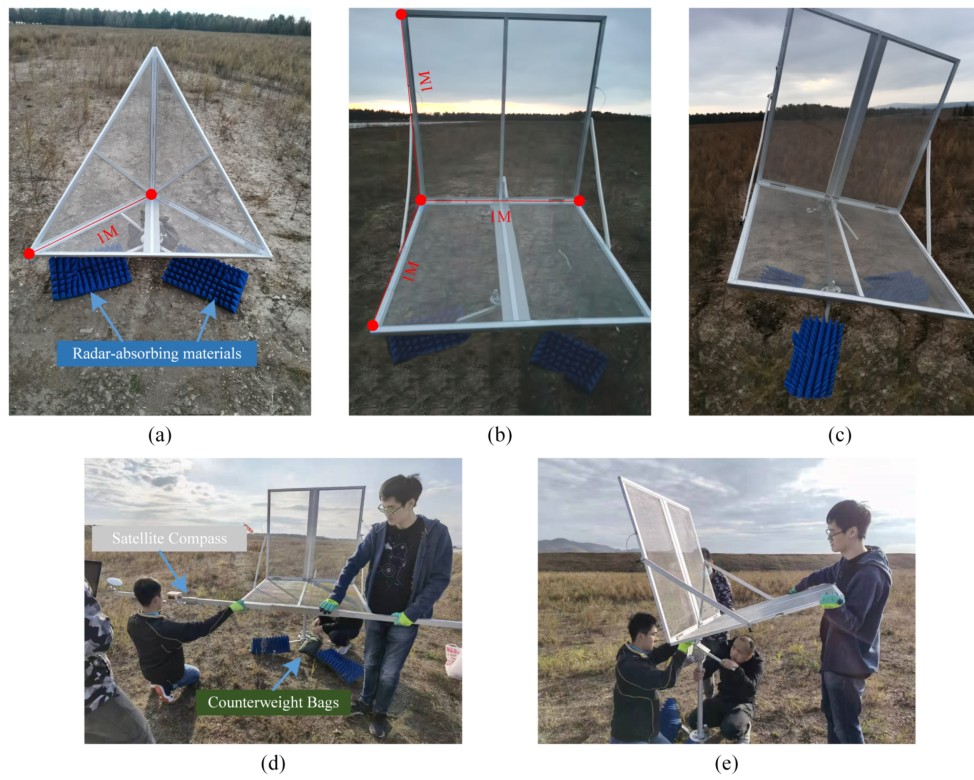

(a)     (b)     (c)

(d)     (e)

**Figure 5.** Ground-deployed CRs at Yudaokou Airport in September 2021: (**a**–**c**) trihedral CR, dihedral CR, and $22.5°$ rotated dihedral CR, respectively; (**d**,**e**) placement process and measuring equipment.

## 4.1. Channel Imbalances Calibration without Additionally Additive Noise

In this experiment, we impose co-pol and x-pol channel imbalances, with linear amplitudes ranging from $-3$ dB to $3$ dB, and linear phases ranging from $-180°$ to $180°$, as well as crosstalk, with random amplitudes in the range $-50$ dB$\sim-15$ dB, and random

phases in the range $-180° \sim 180°$. The crosstalk amplitude is set over a wide range, to verify the robustness of the proposed framework at different levels of crosstalk.

Figure 6a shows the true Pauli RGB image of the CQP SAR data, where the red channel is $\frac{1}{2}|S_{ll} - S_{rr}|^2$, the green channel is $\frac{1}{2}|S_{lr} + S_{rl}|^2$, and the blue channel is $\frac{1}{2}|S_{ll} + S_{rr}|^2$. The areas framed by the red line in Figure 6a are the runway, weedy fields, and steel fence of Yudaokou Airport. Moreover, we placed several CRs on the weedy fields, for different experimental purposes, which are shown in Figure 6b. TR.1–TR.5, DR.1–DR.2, and 225DR.1 indicate the trihedral CRs 1–5, the 0° dihedral CRs, and the 22.5° dihedral CR, respectively. The detailed latitudes, longitudes, and sizes of the CRs are shown in Table 2 and Figure 5a,b According to the flight plan, DR.1–DR.2 and 225DR.1 were prepared for this image. However, the altitude and heading of the route were slightly modified when the airplane took off, which caused the backscatter to deviate slightly from the theoretical values of DR.1–DR.2 and 225DR.1:

$$\left[S_{DR}^{lr}\right] = ([U_{lr}] \otimes [U_{lr}])^T * \begin{bmatrix} 1 & 0 & 0 & -1 \end{bmatrix}^T = \begin{bmatrix} 1 & 0 & 0 & -1 \end{bmatrix}^T \tag{29}$$

$$\left[S_{225DR}^{lr}\right] = ([U_{lr}] \otimes [U_{lr}])^T * \begin{bmatrix} 1 & -1 & -1 & -1 \end{bmatrix}^T = \begin{bmatrix} 1-j & 0 & 0 & -1-j \end{bmatrix}^T \tag{30}$$

**Table 2.** Summery of the CRs. N and E of columns 3 and 4 are north latitude and east longitude, respectively. Detailed schematic diagrams of the CR sizes can be referred to Figure 5a,b.

| CR Types | ID | Longitude (Degree) | Latitude (Degree) | Size (m) |
|---|---|---|---|---|
| | TR.1 | 116.869611E | 42.139556N | 1 |
| | TR.2 | 116.870528E | 42.140233N | 1 |
| Trihedral CR | TR.3 | 116.872422E | 42.138997N | 1 |
| | TR.4 | 116.873844E | 42.138783N | 1 |
| | TR.5 | 116.874867E | 42.139358N | 1 |
| 0° Dihedral CR | DR.1 | 116.871078E | 42.139272N | 1 |
| | DR.2 | 116.873450E | 42.139661N | 1 |
| 22.5° Dihedral CR | 225DR.1 | 116.872117E | 42.140114N | 1 |

Next, we estimate the channel imbalances in the uncalibrated RGB CQP SAR image shown in Figure 6c, which was derived after adding the above crosstalk and channel imbalances. Upon adding crosstalk and channel imbalance, it becomes apparent that the Pauli RGB plot of the simulated uncalibrated image in Figure 6c exhibits significant color variation, when compared to the true Pauli RGB plot of the undistorted image depicted in Figure 6a. This highlights the necessity of calibrating distorted images, in order to obtain precise feature information for practical applications. Figure 6d shows the ENL images derived from Figure 6c. Heterogeneous scenes, such as steel fences, have small ENL values, while homogeneous areas, such as airport runways and bare soil, have larger ENL values; therefore, complex building areas can be removed after setting the ENL threshold, and the remaining parts are essentially homogeneous areas. $R_{vb}$ is shown in Figure 6e. The $R_{vb}$ values of the 0° dihedral CRs and 22.5° dihedral CR are larger than those of the remaining ground objects. The numerical relationships among the $R_{vb}$ values shown in Figure 6e can be deduced directly from (29) and (30). To more intuitively reflect the relative relationship among different ground features, in Figure 6f, the color scale is set to $0 \sim 7$, and there are slightly bright stripes (in the red box) along the range direction: this is because larger crosstalk may impact $R_{vb}$. However, the relative relationship between the volume-dominated and surface-dominated $R_{vb}$ values does not change when the area with larger $R_{vb}$ is dominated by volume scatterers and the area with smaller $R_{vb}$ is dominated by surface scatterers.

Therefore, it is effective to select the area where surface scattering and volume scattering dominate according to the $R_{vb}$ histogram. Moreover, the amplitude and phase filtering operations proposed in this paper are performed when the channel imbalance is initially

estimated, which reduces the error caused by selecting the wrong area. Here, $th_e$ is set as 0.3, and $th_{rl}$ and $th_{ru}$ are set as 0.25 and 0.3, respectively. We note that there is no effective backscatter except noise on the airport runway, so the area selected from the runway cannot be used as effective candidate pixels. When using this selection algorithm to solve the problem, noise-dominated areas, such as calm water and cement floors, should be removed as much as possible.

After setting the number of range and azimuth bins to 100 and 25, respectively, the $\alpha$ and $k$ estimation results are shown in Figure 7. Figure 7a,b represent the preliminary results of the amplitude and phase of $\alpha$ in different azimuth and range bins. By projecting the solution results along the Y axis onto the X-Z plane, we determine that the results of different azimuth bins in the same range bin are approximately the same. According to the $\alpha$ filtering operation shown in Figure 7c,d, the fitting results reduce the error of the initial estimation. After the simulation data are calibrated according to the fitting result of $\alpha$, $k$ is estimated according to the semireflection symmetry, and the results are shown in Figure 7e–h.

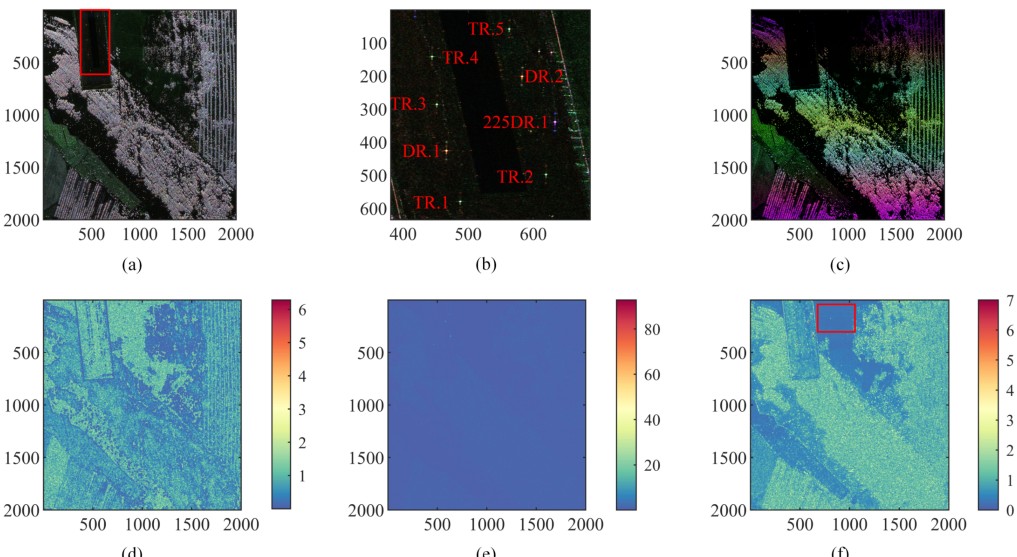

**Figure 6.** L-band CQP SAR images: (**a**) the true Pauli RGB CQP SAR image, where the red channel is $\frac{1}{2}|S_{ll} - S_{rr}|^2$, the green channel is $\frac{1}{2}|S_{lr} + S_{rl}|^2$, and the blue channel is $\frac{1}{2}|S_{ll} + S_{rr}|^2$; the radar illuminated the scene from top to bottom; the airport location is marked in red; (**b**) enlarged view of the near range in (**a**); (**c**) simulated uncalibrated Pauli RGB image; (**d**) ENL image; (**e**,**f**) $R_{vb}$ images with different scales; the red box in (**f**) marks bright stripes.

The fitting results shown in Figure 7g,h indicate that by applying the $k$ filtering operations proposed in this paper, the initially estimated errors shown in Figure 7e,f, due to improper selection of the volume-dominated targets and the presence of low crosstalk, can be reduced. As mentioned above, Figure 7f,h show two ambiguous phases, which can be eliminated by using the DEM. If there is no corresponding DEM for the selected area, the middle-fitting line in the obtained result is selected as the final result. In Figure 7e,f, the control group (CG) results, using (8) and (11) to estimate the amplitude and phase of $k$, are shown in blue lines. Compared to the added channel imbalances, the residual errors of estimating the channel imbalances can be evaluated as

$$\begin{cases} \text{error}_{\text{amp\_dB}} = \text{mean}\left(\left|dB\left(\text{LSL}(\text{amp}_{k/\alpha})/\text{True}(\text{amp}_{k/\alpha})\right)\right|\right) \\ \text{error}_{\text{pha}} = \text{mean}\left(\left|\text{LSL}(\text{pha}_{k/\alpha}) - \text{True}(\text{pha}_{k/\alpha})\right|\right) \end{cases} \tag{31}$$

$$\text{error}_{k/\alpha} = \text{mean}\left(\left|(\text{LSL}(k/\alpha) - \text{True}(k/\alpha))/\text{True}(k/\alpha)\right|\right) \tag{32}$$

where LSL($\cdot$) and True($\cdot$) represent the fitted and true values, respectively.

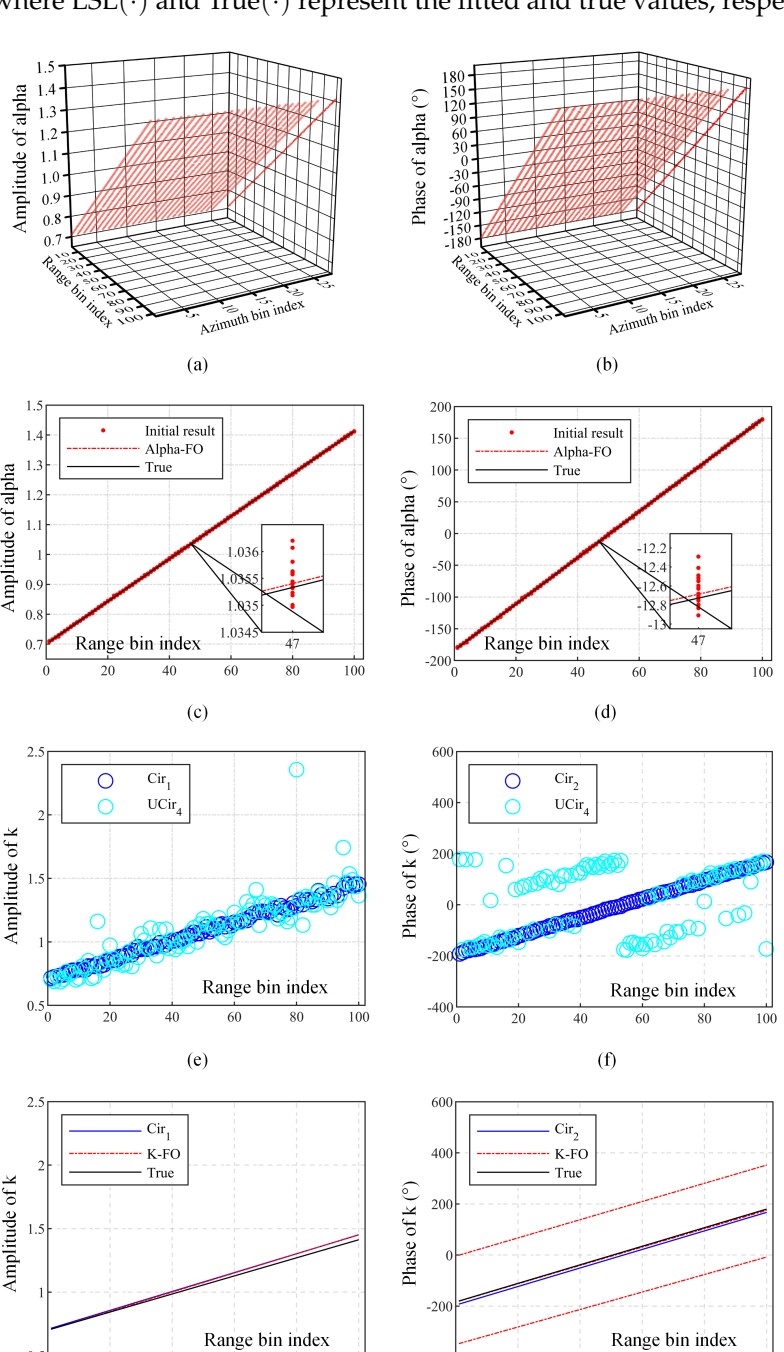

**Figure 7.** Estimated PolCAL parameters of the simulated CQP SAR images: (**a**,**b**) initial amplitude and phase results of $\alpha$; (**c**,**d**) alpha filter amplitude and phase; alpha-FO indicates the $\alpha$ filtering results; (**e**,**f**) initial amplitude and phase results of $k$; (**g**,**h**) K filter amplitude and phase; K-FO denotes the $k$ filtering results after using $Cir_1\&UCir_4/UCir_4$ to estimate the initial amplitude/phase of $k$.

Compared to the added values ('True' in Figure 7c,d) of $\alpha$, the amplitude and phase errors of $\alpha$ with the proposed scheme are $2.4516 \times 10^{-4}$ dB/$0.4865°$, respectively, according to (31); according to (32), error$_\alpha$ is 0.0085. According to (31), comparing the estimated $k$ of the control group ('$Cir_1$' and '$Cir_2$' in Figure 7g,h) with the added values ('True' in Figure 7g,h), the amplitude and phase errors of $k$ are 0.1783 dB and $12.5263°$, respectively, and the values determined by the algorithm proposed in this paper ('K-FO' in Figure 7g,h)

are 0.1458 dB and 2.7558°, respectively. In addition, according to (32), the error$_k$ of the control group is 0.2215, and that of the algorithm proposed in this paper is 0.0515. It should be stressed that the $k$ errors estimated by the proposed scheme and those of the control group may deviate slightly from the true values, because the originally synthesized CQP SAR data may have residual k. Next, the estimated accuracy is further illustrated by point targets, which is the most accurate error estimation method.

In Figure 8, the co-pol and x-pol signatures of the strong point targets are plotted on a linear polarimetric basis. The original signature diagrams are presented in rows 1 and 5. After adding the channel imbalances and crosstalk, the polarization signatures are distorted, as shown in rows 2 and 6. The results of the proposed scheme and the control group are presented in rows 3 and 7, and in rows 4 and 8, respectively. Compared to rows 1 and 5, the control group results have apparent deviations (marked by the dark red circles in Figure 8), while the results of the proposed scheme do not.

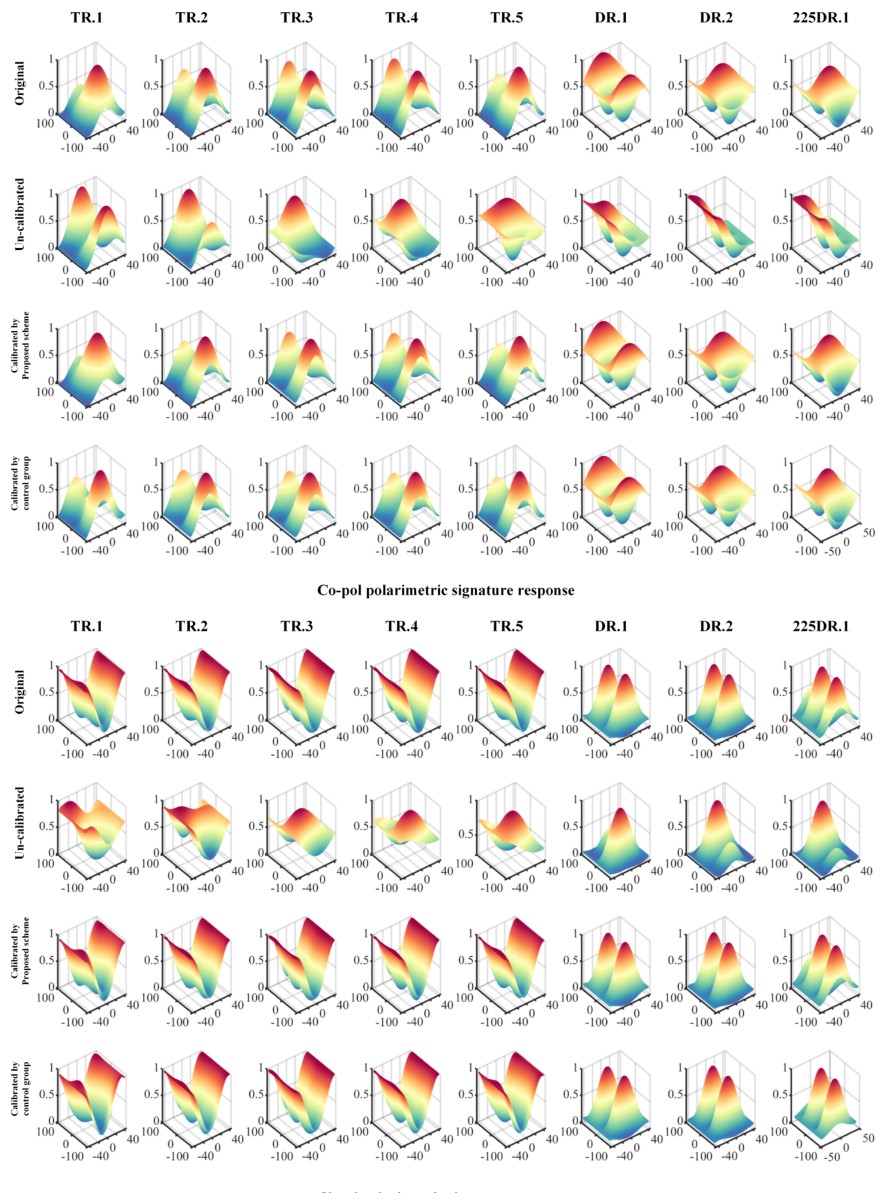

**Figure 8.** Polarimetric (co-pol and x-pol) signature responses of the strong point targets in the CQP SAR data set, without additive noise in linear polarimetric basis.

A quantitative evaluation of DR.1–DR.2 and 225DR.1 is presented in Table 3. Columns 1–2, 3–4, and 5–6 show the amplitude ratio (dB) and phase difference (degrees) between the LL and RR in the original synthesized CQP SAR images, the CQP SAR images calibrated by the proposed scheme, and the CQP SAR images calibrated by the control group, respectively. Due to the change in the flight route after deploying the CRs, the values presented in columns 1–2 are slightly different from the theoretical values. According to the evaluation system shown in (31), the mean residual errors of the three CRs (DR.1–DR.2 and 225DR.1) are 0.164 dB/2.590°, with the proposed scheme, and 0.292 dB/25.369° for the control group. Therefore, the proposed scheme is a more feasible calibration choice than the control group.

**Table 3.** Quality evaluation of DR.1–DR.2 and 225DR.1 in the CQP SAR data set without additionally additive noise. The unit of measure in the second row, the fourth row, and the fifth row is dBs, and the unit of measure in the third row is degrees (°). For obvious comparison, the values in brackets in the seventh column are in phase periods of 0 to 360 degrees. The last row shows the errors of the results of the proposed method and the control group compared to the corner reflectors by (31).

| | Original | | Proposed Scheme | | Control Group | |
|---|---|---|---|---|---|---|
| DR.1 | 0.923 | 172.029 | 0.902 | 174.946 | 0.890 | −168.993 (191.007) |
| DR.2 | 0.932 | 172.224 | 0.933 | 174.576 | 0.917 | −160.542 (199.458) |
| 225DR.1 | 0.911 | 90.195 | 0.882 | 92.696 | 0.869 | 120.089 |
| Mean error by (31) | - | | 0.164 | 2.590 | 0.292 | 25.369 |

### 4.2. Channel Imbalances Calibration with Additionally Additive Noise

Considering that the additional noise in the SAR images varies, we append random additional noise to the simulated uncalibrated CQP SAR images used in Section 4.1 to validate the accuracy and robustness of the proposed scheme according to (2). The ratios of the additional noise amplitudes in each channel to the span are within $-60$ dB$\sim-25$ dB, and the noise phases are within $-180°\sim180°$. In Figure 9, the co-pol and x-pol signatures of the CRs are plotted, where the meaning of each row is the same as in Figure 8. Although the additional noise affects the CR signatures, the results of the proposed scheme are still similar to the original results, and perform better than the control group. A quantitative evaluation of DR.1–DR.2 and 225DR.1, with additional noise, is presented in Table 4.

**Table 4.** Quality evaluation of DR.1–DR.2 and 225DR.1 in CQP SAR data set with additionally additive noise. The unit of measure in the second row, the fourth row, and the fifth row is dBs, and the unit of measure in the third row is degrees (°). For obvious comparison, the values in brackets in the seventh column are in phase periods of 0 to 360 degrees. The last row shows the errors of the results of the proposed method and the control group compared to the corner reflectors by (31).

| | Original | | Proposed Scheme | | Control Group | |
|---|---|---|---|---|---|---|
| DR.1 | 0.923 | 172.029 | 0.943 | 178.953 | 0.955 | −164.271 (195.727) |
| DR.2 | 0.932 | 172.224 | 0.983 | 178.351 | 1.000 | −158.684 (201.316) |
| 225DR.1 | 0.911 | 90.195 | 0.999 | 93.755 | 1.014 | 119.529 |
| Mean error by (31) | - | | 0.614 | 4.154 | 0.927 | 23.893 |

Overall, the additional noise impacts the statistical properties of the distributed targets, resulting in larger estimated errors than when additional noise is not included, in most cases. As shown in Table 2, the average amplitude and phase errors of the proposed scheme and control group are 0.614 dB/4.154° and 0.927 dB/23.893°, respectively. Combined with the results in Section 4.1, these results show that the proposed scheme is a better calibration choice for channel imbalances with distributed targets in CQP SAR images with reciprocal crosstalk.

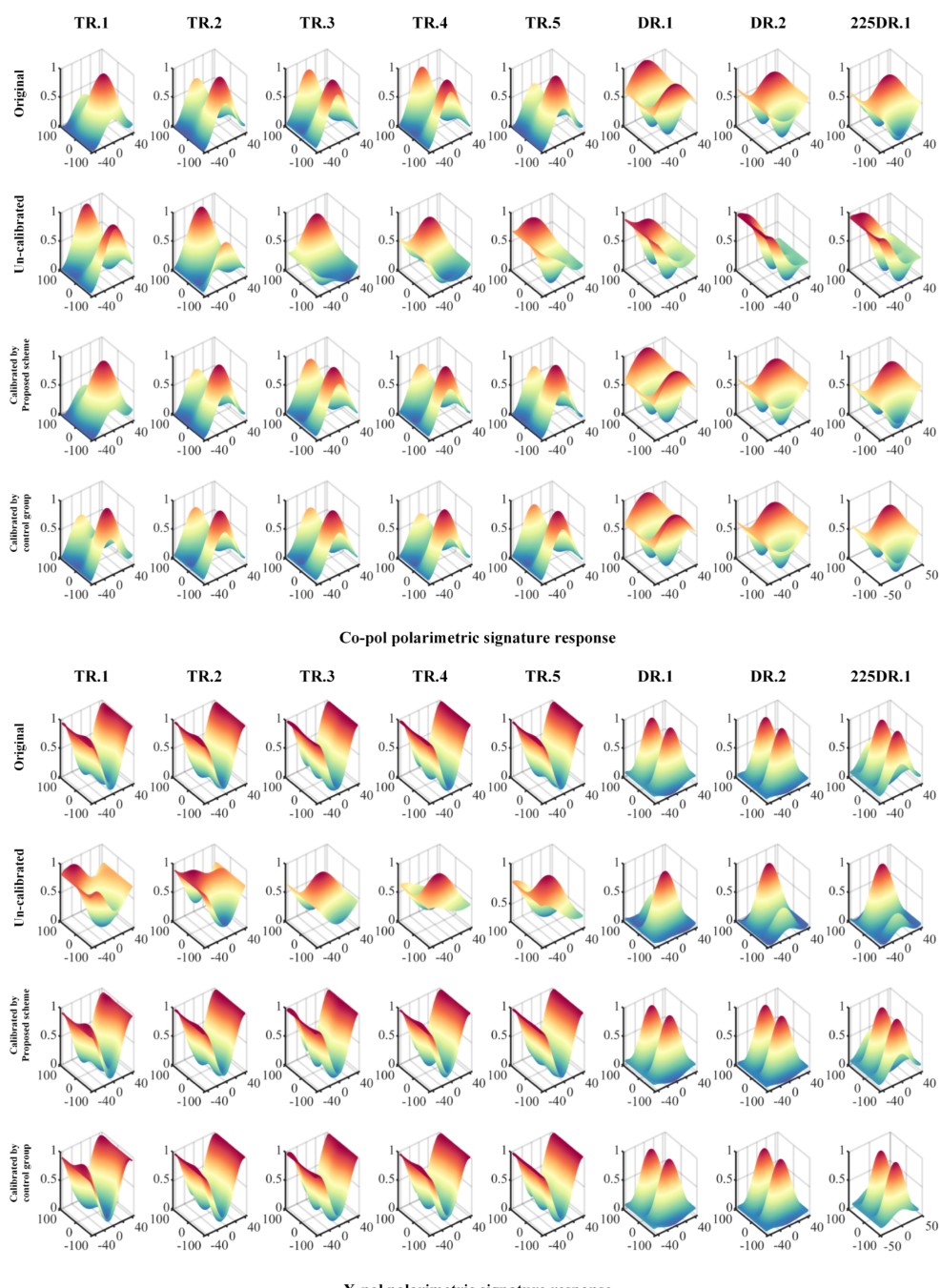

**Figure 9.** Polarimetric (co-pol and x-pol) signature responses of the strong point targets in the CQP SAR data set, with additive noise on linear polarimetric basis.

## 5. Discussion

The above algorithm includes some factors, such as threshold selection, that may affect accuracy. In addition, the effect of crosstalk on estimating channel imbalances should be further discussed, in order to design SAR systems. The detailed discussions of these problems are presented in this section.

### 5.1. Number of Range and Azimuth Bins

In this article, the CQP SAR images are divided into different bins along the azimuth and range directions, when estimating the channel imbalances. The number of range bins is strongly related to the drift in the range direction in CQP SAR systems. When the variation

in the channel imbalance along the range direction is stable, less range bins are needed. The number of azimuth bins is highly dependent on the Newton iteration calculation process in (22). Because the unknowns in (22) are the real and imaginary parts of $p$, the number of azimuth bins, $D$, is set to greater than or equal to 2. When $D$ is set to be larger than 2, the iterative process may be more accurate, but the computational efficiency is reduced. If computational efficiency is the main priority, the readers may prefer to try a smaller $D$, and compare the corresponding estimated results with the CR results, to determine the value that meets the design specification, before calibrating the other images. If CRs are unavailable, it is recommended to set a larger $D$, to ensure high estimation accuracy. It is also possible to combine the results estimated under multiple bins, and to filter the results to obtain high estimation accuracy [22].

### 5.2. Choice of ENL Threshold

When extracting surface-dominated and volume-dominated targets before PolCAL, the ENL is utilized to remove heterogeneous scattering areas, such as cities. The heterogeneous scattering areas are removed for the following reasons: (1) when using $R_{vb}$ to extract volume-dominated areas, targets dominated by secondary scattering are also selected from heterogeneous areas, such as cities, resulting in reduced calibration accuracy when the semireflection symmetry is applied; if the ENL is used to remove as many secondary scattering areas as possible, the accuracy of the selected areas is effectively increased, increasing the accuracy of the final calibration result to some extent; (2) in practical applications, we usually use the multilook operation to estimate the covariance matrix; at the boundary between heterogeneous and homogeneous targets, the multilook operation may make the $R_{vb}$ values of the extracted features differ from those of the required features; as a result, heterogeneous scattering features may be extracted, decreasing the PolCAL accuracy. For example, mixing urban targets and dark targets increases $R_{vb}$, as the targets may be misinterpreted as volume scattering targets. The heterogeneous property leads to a small ENL for complex scattering areas, such as cities, and a high ENL for naturally homogeneous areas; therefore, urban areas can be eliminated by setting an ENL threshold. Here, we set the ENL threshold to 0.3, to remove complex scattering areas, and select more natural areas dominated by surface scattering and volume scattering.

### 5.3. Choice of $R_{vb}$ Threshold

After removing complex scattering regions, such as cities, the selection of appropriate scattering properties for channel imbalance calibration remains a great challenge, as many polarization parameters vary with system distortion. In addition to the time-consuming and laborious manual selection, in this paper, $R_{vb}$ is considered an available option to extract surface-dominated and volume-dominated targets in uncalibrated CQP SAR images. In Figure 10, we show the scatter characteristics of our extracted targets in the $H/\alpha$ plane [36], where $th_e$, $th_{ru}$, and $th_{rl}$ are set to 0.3, 0.3, and 0.25, respectively.

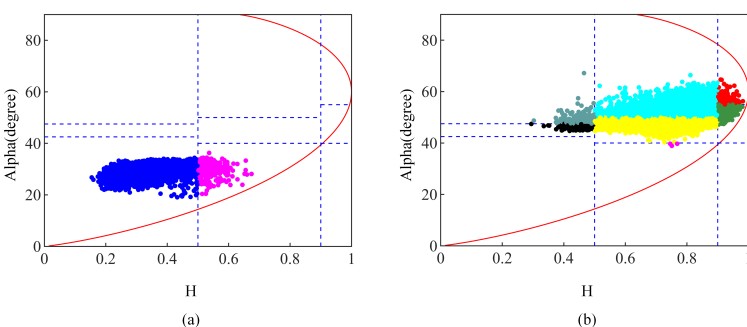

**Figure 10.** The (**a**) surface-dominated and (**b**) volume-dominated targets extracted from the uncalibrated CQP SAR data in Section 4.1. Different point colors represent different areas in the $H/\alpha$ plane.

In Figure 10, most surface-dominated points are found at low-entropy, medium-entropy, and low-alpha angles, while volume-dominated points are found at medium-entropy, high-entropy, low-alpha, and medium-alpha angles. However, the selected area, which should be dominated by volume scattering, also contains a smaller number of surface-dominated targets. This is because most of the polarization parameters cannot accurately represent the characteristics of the selected area, due to system distortion. In fact, there are very few algorithms for selecting points with absolutely correct scattering properties in uncalibrated CQP SAR data. It should be stressed that the accuracy in this paper depends not only on the quantity and quality of the selected regions, but also on the channel imbalance filtering operations. After the channel imbalances are initially estimated, the results obtained, based on the inaccurate scattering areas, are filtered out as much as possible, thereby improving the final calibration results.

In Figure 11, we evaluate the errors of estimating the channel imbalances based on (31) by setting different $R_{vb}$ thresholds. In the following section, we set ENL to 0.3, while $th_{rl}$ and $th_{ru}$ are uniformly changed from 0.2 to 0.8 in 0.01 steps.

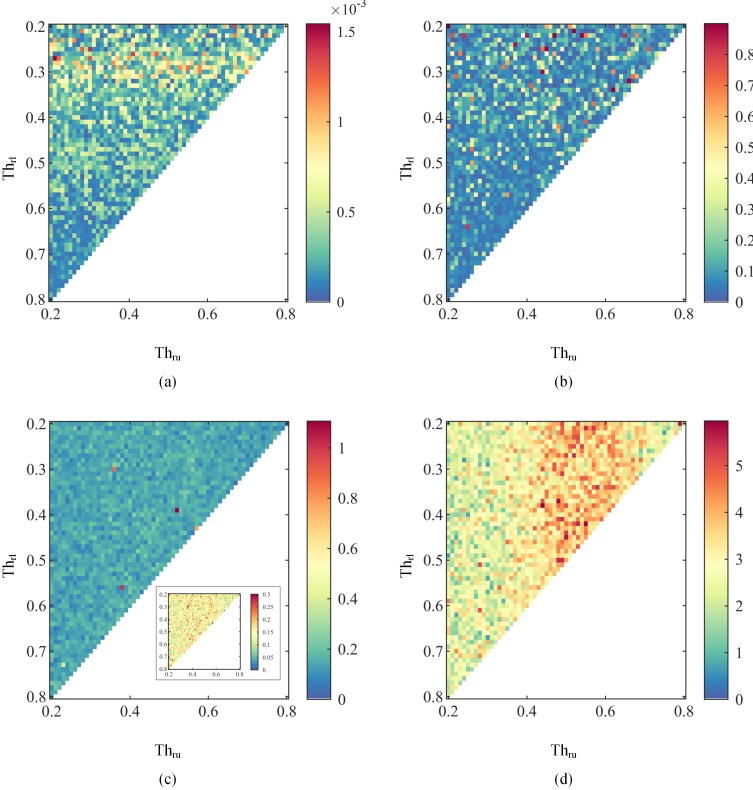

**Figure 11.** Calibration accuracy of the proposed scheme under different $R_{vb}$ thresholds: (**a**,**b**) amplitude and phase errors of $\alpha$; (**c**,**d**) amplitude and phase errors of $k$.

As previously mentioned, in the proposed scheme, unknown reciprocal crosstalk does not significantly impact the estimation of $\alpha$ when the CQP SAR image is reciprocal. In Figure 11a,b, when different $th_{rl}$ are set to select the surface-dominated targets, the amplitude and phase errors of $\alpha$ are relatively small, within $1.5 \times 10^{-3}$ dB/0.8°. Even if the volume-dominated targets are selected to estimate $\alpha$, the calibration accuracy of $\alpha$ is ensured by the reciprocity of the volume-dominated targets. In practical PolCAL, when $th_{rl}$ is set to a larger value, the computational time to estimate $\alpha$ increases, which may not be acceptable in multiple image calibration. Therefore, a lower $th_{rl}$ value, such as 0.25, is recommended to improve the calibration efficiency when ensuring the calibration accuracy.

The estimated errors of $k$ are presented in Figure 11c,d. In the amplitude errors of Figure 11c, when the wrong semireflection symmetry area is selected, by setting the inappropriate thresholds, the fitting operations will increase the instability of the estimated

results, which will lead to a worse estimated result than other situations. However, most of the amplitude errors are stable, below 0.4 dB.

In Figure 11d, as $th_{ru}$ increases, the estimated phase errors gradually increase up to a certain point, and then start to decrease. The same situation is shown in the reset maximum thumbnail of Figure 11c. As $th_{ru}$ increases, the energy of the selected targets in the co-pol channel is smaller than that in the volume-dominated area, which causes noise to affect the semireflection symmetry of the selected area, and increases the estimation errors: this is because, as an increasing number of surface-dominated targets are selected that satisfy the semireflection symmetry, the solution errors decrease according to the filter operation. We note that when more semireflective symmetry targets are applied to estimate $k$, the noise effect on the accuracy calibration of $k$ is attenuated, although noise in the image still affects the semireflection symmetry in the surface-dominated regions. There is a combined area between the surface-dominated and volume-dominated regions, and the reflection symmetry of this area was not analyzed in detail, which seriously affects the accuracy of $k$, as shown in Figure 11c; therefore, we select only volume-scattering-dominant regions, and 0.3 is recommended as the optimal value of $th_{ru}$.

*5.4. Crosstalk Effect on Channel Imbalance Estimation*

In Section 4, the simulated uncalibrated CQP SAR data are used to verify the capability of the proposed scheme, when the reciprocal crosstalk along the range direction is randomly set within a wide range of $-50$ dB$\sim-15$ dB, to verify the adaptability of different crosstalk systems: namely, $-50$ dB $\leq |u| = |z|, |v| = |w| \leq -15$ dB and $-180° \leq \angle u = \angle z, \angle v = \angle w \leq 180°$. When researchers design CQP SAR systems, the crosstalk (i.e., the power leakage between channels) between the antennas is usually verified by a fixed value, for example, better than $-35$ dB, better than $-30$ dB, etc; therefore, in this part, different levels of fixed crosstalk are added to the undistorted CQP SAR images, to estimate the channel imbalance, and to calculate the determination error, with the aim of validating the proposed framework as an effective tool for system design.

In each trial, we impose: (1) system crosstalk with constant amplitudes ranging from $-50$ dB to $-15$ dB ($-50$ dB $\leq |u| = |z| = |v| = |w| \leq -15$ dB) and random phases ($-180° \leq \angle u = \angle z = \angle v = \angle w \leq 180°$); (2) channel imbalances with linear amplitudes ranging from $-3$ dB to 3 dB and linear phases ranging from $-180°$ to $180°$; $th_e$, $th_{ru}$ and $th_{rl}$ are set to 0.3, 0.3, and 0.25, respectively. To show the effect of the crosstalk phase on the channel imbalance estimation results, 45 Monte Carlo trials were conducted for each crosstalk amplitude. The results are shown in the boxplots in Figure 12.

As has been emphasized throughout the article, reciprocal crosstalk has little negative effect on the $\alpha$ estimation accuracy. In Figure 11a,b, as the crosstalk increases, the $\alpha$ estimation error remains low, within $1.4 \times 10^{-3}$ dB and $0.7°$. However, the crosstalk affects $R_{vb}$, the selection of the volume-scattering-dominant region, and the establishment of (22). Thus, in Figure 12c,d, different crosstalk unknowns have more pronounced impacts on estimating $k$ than $\alpha$. As the crosstalk amplitude increases, the accuracy and stability of estimating $k$ decreases significantly. When the crosstalk amplitude is greater than $-18$ dB, the corresponding estimation accuracy drops directly from less than 0.4 dB to more than 0.5 dB. In addition, the stability effect of the crosstalk phase on the amplitude estimation becomes obvious, and the cabinet length of $-17$ dB is nearly twice that of $-18$ dB. If 0.5 dB/5° (such as GF-3) is used as the channel imbalance design index, then this algorithm should be used with crosstalk greater than $-23$ dB.

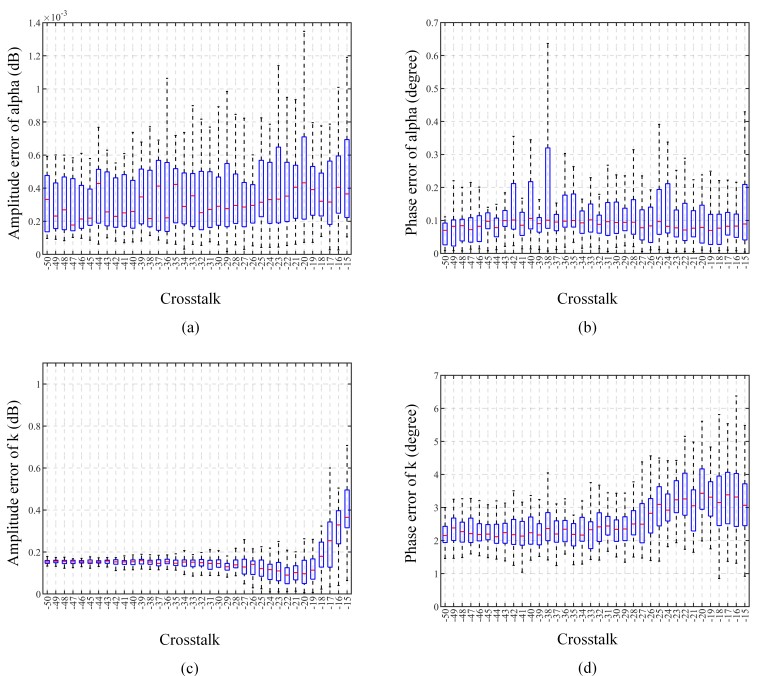

**Figure 12.** Effect of different crosstalk levels on the channel imbalance estimation: (**a**,**b**) amplitude and phase errors of $\alpha$; (**c**,**d**) amplitude and phase errors of $k$. The standard boxplot notation is used (lower/upper hinges—first/third quartiles; whiskers extend from the hinges to the largest/lowest values no further than $1.5 \times$ interquartile ranges).

## 6. Conclusions

In the existing distributed target calibration for channel imbalances of CQP SAR systems with reciprocal crosstalk, some reflection symmetry assumptions of volume-dominated targets are used to estimate the phase of the co-pol channel imbalance $k$, i.e., (9)–(11). This paper notes that the reflection symmetry used to estimate the phase of $k$ has poor applicability in volume-dominated target regions, which reduces the estimation accuracy of the $k$ phase. In this paper, we assess the applicability of the reflection symmetry in volume-dominated target regions, and propose an algorithm for $k$ phase calibration based on high-priority reflection symmetry (we name it semireflection symmetry) in volume-dominated targets. Furthermore, as the reciprocity of the surface-dominated targets is sufficient to estimate the x-pol channel imbalance $\alpha$ with [20], a channel imbalance calibration scheme based on crosstalk reciprocity in SAR systems is proposed for polarimetric applications, in which crosstalk can be ignored and CRs are not applied to calibrate CQP SAR images. The framework includes the automatic extraction of surface-dominated and volume-dominated targets as reference pixels, the estimation and filtering of $\alpha$ in the surface-dominated target regions, and the estimation and filtering of $k$ in the volume-dominated target regions. The filtering operations aim to obtain more robust results. We conducted experiments based on CQP SAR images synthesized from reciprocal L-band LQP SAR images. A high calibration accuracy of $\alpha$ was achieved by the experimental part, which was consistent with [20]. For the $k$ calibration, compared to the control group, the algorithm proposed in this paper has an improvement of 0.1 dB/22° by CRs without additionally additive noise. The improvement can reach 0.3 dB/19° with additionally additive noise. It is recommended that $th_e$, $th_{ru}$ and $th_{rl}$ are set to 0.3, 0.3, and 0.25, respectively, to select the appropriate surface-dominated and volume-dominated targets. In addition, we recommend that researchers use this method when the crosstalk design specification of the system is better than $-23$ dB. In future work, we will implement the scheme in more measured CQP SAR images, to improve the method.

**Author Contributions:** Conceptualization, Y.D. and X.L.; methodology, X.Z.; software, X.Z.; validation, X.Z.; formal analysis, H.Z.; investigation, X.Z. and H.Z.; resources, H.Z., Y.D. and X.L.; writing—original draft preparation, X.Z.; writing—review and editing, X.Z., H.Z., Y.D. and X.L.; project administration, Y.D. and X.L.; funding acquisition, Y.D. and X.L. All authors have read and agreed to the published version of the manuscript.

**Funding:** This research was funded by the Beijing Municipal Natural Science Foundation (grant no. 4192065) and the National Natural Science Foundation of China (grant no. 61901445).

**Data Availability Statement:** Not applicable.

**Acknowledgments:** The authors would like to thank the Department of Space Microwave Remote Sensing System, Aerospace Information Research Institute, Chinese Academy of Sciences for providing L-band SAR data. The authors would also like to thank all colleagues who participated in this experiment.

**Conflicts of Interest:** The authors declare no conflict of interest.

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
