# Peer review of "A Channel Imbalance Calibration Scheme with Distributed Targets for Circular Quad-Polarization SAR with Reciprocal Crosstalk"

_remotesensing, doi:10.3390/rs15051365_

Round 1
Reviewer 1 Report
The research paper highlights its innovative approach to channel imbalance calibration in CQP SAR. The proposed method reduces the need for corner reflectors in channel imbalance calibration of CQP SAR when crosstalk unknowns are small and reciprocity is present, making it a noteworthy contribution to the field. The proposed semireflection symmetry assumptions for volume-dominated targets enhances the accuracy of the co-polarization channel imbalance phase. The paper is well-structured and effectively conveys its novelty and motivation. The reviewer supports its publication with minor revisions to ensure its readability and clarity.
1. Some of the formats used in English should be modified to improve the readability of the paper:
1) Page 2 Line 61-Line 63: The corresponding left bracket should be added to make the article more readable.
2) Page 2 Line 85: There may be a better place to put the reference [20] citation. This version now makes it less readable in the words where it is located.
3) Title of Figure 1: There should be a comma between ‘ACir2’ and ‘and’.
4) Page 11 Line 381: Please correct ‘at the same distance’.
5) Figure 7: Mark the units of (b), (d), (f), and (h).
6) Page21 Line 614: Check thanalpha.
2. Page11 Line 353: What is the first d iteration? Does it mean that the iteration is done only once?
3. Spatial averaging is a basic step in this paper. In the experimental section, the authors lack a description of the spatial averaging.
4. In Tables 1 and 2 of Section 4, the authors give more precise calibration accuracy using angular reflectors and by comparison with existing methods. However, this approach is not intuitive and the average error values are more indicative of the advantages of the proposed method.
Reviewer 2 Report
This paper proposes a channel imbalance calibration method for a circular quad-polarization SAR (hereafter CQP SAR) using distributed reflectors. The proposal is interesting and the experimental results shows fine. On the other hand, some descriptions are unclear.
1. First of all, the definitions in Eq.1 is different from the cited article [20]. The authors wrote “… where the subscripts l/r indicate left-handed/right-handed transmit/receive channels…” However, in [20], it is written “… antenna P will transmit polarization p with scaling factor t^{a}_{pp} and polarization q with scaling factor t^{a}_{qp} (ideally zero).” That is, the definitions of the subscripts are different from each other. t_{ll} and t_{rl} are left- and right-handed signals transmitted from an identical “l” antenna simultaneously in [20] while they are left- and right-handed transmitted signals received by an identical “l” antenna. I don’t understand why the experiments in this article worked properly though the mathematical understandings are completely different. Please throughout check the equations, descriptions and experiments. It is better to describe the assumed system more precisely instead of just citing [20].
2. In Section 4, experimental setups should be clearly described. There is no parameter for SAR. The authors only wrote it is “L-band” and no information for, for example, center frequency, bandwidth, PRF, altitude, speed etc. There is no summary for the reflectors.
3. Scales are not shown in the figures. For example, in Fig. 6. there is no power scale in (a), no descriptions for the meaning of the color in (c),
Reviewer 3 Report
1. The aim of this study, assesing priority of reflection symmetry properties of volume dominated targets used to calibrate copol channel imbalance phase in CQP SAR data synthesised from linear quad-polarization data of ALOS, GF-3, and RADARSAT-2.
2. I think the topic original or relevant in the field of remote sensing
3. According to this paper notes that the reflection symmetry used to estimate the phase of k has poor applicability in volume-dominated target regions, which reduces the estimation accuracy of the k phase
4.
-What is SAR system parameters? please mention it at introduction sectin
-What is distortion? what are distortion parameters. please explain
-According to paper the filtering operations can be divided into three steps: how did you calibrate the covariance matrices and why do you need
5. Please read this paper for introduction :Duysak, H. & YiÄŸit, E. (2022). Investigation of the performance of different wavelet-based fusions of SAR and optical images using Sentinel-1 and Sentinel-2 datasets . International Journal of Engineering and Geosciences , 7 (1) , 81-90 . DOI: 10.26833/ijeg.882589
7. Please include any additional comments on the tables and figures.
Fig. 5 is not clear please enlarge it
Author Response
Dear Reviewer3,
We are writing to respond to your comments on our manuscript, "A Channel Imbalance Calibration Scheme with Distributed Targets for Circular Quad-Polarization SAR with Reciprocal Crosstalk", whose ID is remotesensing-2235716. We appreciate your taking the time to review our work and providing us with valuable feedback. Please find attached our response letter <author-coverletter-27222003.v1.pdf>.
We would like to address a point that may have caused confusion. We noticed that your reviewer alias changed from "Reviewer2" to "Reviewer3" after we submitted our response to your comments. Please rest assured that we understand that you are the same reviewer who provided us with constructive feedback. We apologize for any confusion this may have caused.
We appreciate your insightful comments, and we have carefully considered each of your suggestions in our revision. We believe that our revised manuscript has addressed your concerns and improved the clarity and quality of our work.
Thank you again for taking the time and effort to review our manuscript. Please let us know if you have any further comments or concerns.
Best regards
The authors
